# Neuromuscular Response Disparities in Non-Professional Athletes during Side-Cutting: Exploring Sex Differences through Electromyographic Analysis

Adrián Feria-Madueño [1], Jose A. Parraca [2,3], Nuno Batalha [2,3] and Borja Sañudo [1,*]

[1]  Department of Physical Education and Sport, University of Seville, 41004 Seville, Spain; aferia1@us.es
[2]  Departamento de Desporto e Saúde, Escola de Saúde e Desenvolvimento Humano, Universidade de Évora, 7004-516 Évora, Portugal; jparraca@uevora.pt (J.A.P.); nmpba@uevora.pt (N.B.)
[3]  Comprehensive Health Research Centre (CHRC), University of Évora, 7004-516 Évora, Portugal
*  Correspondence: bsancor@us.es

**Abstract:** This study aims to fill a knowledge gap by investigating electromyographic disparities in anterior and posterior muscle activation and coactivation ratios among non-professional men and women during side-cuttings. A cohort of 162 non-professional athletes participated in directional change maneuvers. Electromyographic assessments focused on coactivation ratios during the initial 50, 100, 150, and 200 ms of contraction, analyzing thigh muscle activations and exploring sex-based differences. Findings revealed higher quadriceps than hamstring muscle activation during directional changes, emphasizing the pivotal role of timing and coactivation ratios. Although the coactivation ratio, indicative of protective muscle control, approached 1 in all directional changes, 40% of subjects exhibited ratios below 0.8, suggesting an elevated injury risk. During open side-cutting at 30°, no significant sex differences were observed in anterior and posterior thigh muscle activation. However, in explosive ratios, women outperformed men, potentially attributable to uncontrolled motor unit recruitment. In open side-cutting at 45° and closed side-cutting at 45°, women displayed significantly higher H/Q ratios, indicating a nuanced sex-specific response. The study underscores the importance of an innovative coactivation ratio approach, revealing its early association with injury risk. Although anterior thigh muscle activation generally exceeded posterior, women exhibited poorer coactivation, potentially heightening knee injury risks during directional changes. This research contributes valuable insights into neuromuscular responses among non-professional athletes, particularly within the context of sex-specific differences.

**Keywords:** side-cutting; neuromuscular; sex differences; knee injuries



## 1. Introduction

Sex considerations emerge as pivotal when evaluating the occurrence and prevention of knee injuries during directional change maneuvers, as highlighted in prior research [1–3]. Among the anatomical structures vulnerable in this context is the anterior cruciate ligament (ACL). Previous studies have consistently shown that women are at a heightened risk of ACL injuries compared to men, especially in sports characterized by frequent directional changes and deceleration [4].

Although kinetic and kinematic factors may be linked to an increased risk of knee injuries [5–7], recent research has highlighted the importance of neuromuscular factors [8,9] in this context. Recent studies have shed light on the impact of training on neuromuscular response during directional change maneuvers [10,11].

The assessment of neuromuscular response has been conducted using various methodologies, with electromyography being an effective tool to examine such responses in specific sports situations [12,13]. Electromyographic research has revealed a reduction in muscle activity in the thigh muscles of women compared to men during directional changes [4]. This

decrease in muscle activity may contribute to knee instability and, consequently, increase the risk of injuries [14]. Although muscle activity has been linked to the risk of injuries, it appears that the relationship between anterior and posterior muscle activity in the lower limbs, expressed through the ratio at the time of directional change, plays a crucial role [9].

Previous studies have considered the analysis of the co-activation ratio in relation to knee stability [15], providing results in percentages. In this way, Agaard et al. determined that the activation of the hamstring muscles relative to the quadriceps was between 15 and 35%. Similarly, researchers [13] have examined the pre-activation of the hamstring muscles, specifically the biceps femoris and semitendinosus muscles, evaluating them in the 10 ms immediately before foot contact with the ground in open-direction changes. The data revealed that the biceps femoris was activated at 30% of MVC, while the activation of the semitendinosus was at 41% of MVC. Specifically, reduced pre-activity of the semitendinosus, leading to a poorer co-activation ratio, appears to be a determining factor in the increased risk of knee injury [16]. These authors found a 30% reduction in the risk of injury when the pre-activity of the semitendinosus increased by 10% during direction changes. In their analysis, the anterior thigh muscles were activated between 79 and 84%, while the posterior muscles were activated between 65 and 68%.

Despite finding data expressed in percentages, authors such as Kristensen et al. did provide numerical data on the ratio evaluated during static balance tasks. For two-legged exercises with open eyes, the ratio was 1.21, whereas for two-legged exercises with closed eyes, the co-activation ratio was 1.43 [17]. Monopodal exercises exhibited elevated ratios, with a co-activation of 2.98 observed under open-eye conditions. However, it is important to note that the tests conducted in this study were not dynamic. The literature reports co-activation ratios during directional changes, differentiating between men and women [18]. In this study, 40 NCAA Division I varsity soccer players were assessed for the hamstring-to-quadriceps (H/Q) ratio at two different moments during open directional changes of 60°: in the 50 ms preceding ground contact and in a loading phase, defined as the initial 50% of the stance phase during the side-step cutting maneuver. In the preparatory phase, women exhibited average ratios of 1.16, whereas men showed ratios of 0.81. In the loading phase, women demonstrated ratios of 1.55, whereas men had ratios of 1.08.

Despite existing evidence on sex differences in neuromuscular responses, there has been a lack of research regarding electromyographic differences related to the activation of anterior and posterior muscles, as well as the ratio between non-professional men and women during different directional changes. In this context, the present study hypothesizes that women will show lower muscle activation and deficits in H/Q coactivation ratios compared to men during different directional change maneuvers. Therefore, the main objective of this study is to carry out a comparison of neuromuscular responses, evaluated through EMG, between men and women during different directional change maneuvers.

## 2. Materials and Methods

### 2.1. Subjects

Two hundred non-professional athletes were recruited for the present study. Following the initial meeting, inclusion criteria were explained. These criteria involved individuals regularly engaging in non-professional physical activity—specifically, performing physical activity for a minimum of 30 min a day at least 3 days per week—and not having suffered a knee or ankle injury in the 12 months preceding the assessment. After this meeting, 20 athletes withdrew from the study, resulting in a total sample of 162 participants (75% men, 25% women). All participants were surveyed to ascertain their weekly and daily physical activity, as well as their height and body mass (Table 1). Participants were verbally briefed on the potential risks of the tests and provided informed consent. The study received approval from the Ethics Committee of the University of Seville.

**Table 1.** Descriptive characteristics of the sample.

| Variables | N = 162 | Men | Women | *p* |
|---|---|---|---|---|
| Sex | | 122 | 40 | |
| Age (years) | 24 (±3) | 25 (±2) | 22 (±1) | 0.110 |
| Body mass (Kg) | 72.84 (±12.76) | 76.92 (±8.12) | 65.36 (±4.52) | 0.103 |
| Hight (m) | 1.74 (±0.07) | 1.80 (±0.09) | 1.63 (±0.06) | 0.071 |
| Body mass index (Kg/m$^2$) | 23.78 (±2.86) | 24.75 (±1.12) | 21.34 (±2.12) | 0.096 |
| Physical activity per week (h/week) | 8.38 (±4.01) | 8.79 (±3.96) | 7.91 (±2.34) | 0.101 |

Data expressed as mean and SD. $p \leq 0.05$. Kg = kilograms; m = meters; Kg/m$^2$ = kilograms divided by meters squared; h/week = hours per week.

### 2.2. Procedure

Participants underwent a maximal voluntary contraction (MVC) test (Biodex Multijoint System, Shirley, New York). Subsequently, subjects performed the directional change test, consisting of two open cuts at 45° (SC45) and 30° (SC30) and one closed cut (SC45cl) on a force platform (Kistler 9260 AA6, Winterthur, Switzerland). Prior to the test, all participants engaged in a standardized warm-up, consisting of cycling on a cycloergometer for 5 min (Ergoline 900, Ergometrics, Bitz, Germany) at 60 W (60 rpm), and familiarization with sports-specific side-cutting maneuvers (5 to 8 attempts). The velocity was controlled by a metronome, set at 4–5.5 m*s$^{-1}$, and the direction of the movements was marked on the floor with tape.

### 2.3. Electromyography

Electromyographic activity of the anterior and posterior thigh musculature (bioPLUX, Lisbon, Portugal) was assessed following the protocol described by the European Project "Surface ElectroMyoGraphy for the Non-Invasive Assessment of Muscles" (SENIAM: http://www.seniam.org, accessed on 22 February 2024). Prior to isometric contraction, the rectus femoris and biceps femoris of the dominant leg were palpated for electrode placement (self-adhesive electrodes with a diameter of 1.5 cm, distance between electrodes 3 cm; Blue Sensor, Medicotest A/S, Olstykke, Denmark). The reference electrode was placed on the patella. During the MVC test, signals were recorded to the maximum value of the root mean square (RMS)-EMG and during side-cuttings at 50, 100, 150, and 200 milliseconds immediately after initial foot contact.

Data on the activation of both the anterior and posterior thigh musculature of each subject were obtained. All values were analyzed 50, 100, 150, and 200 ms before the peak activation of the musculature and in comparison with the MVC for each movement.

### 2.4. Statistical Analysis

Statistical analysis was conducted using the SPSS 22.0 software package (SPSS Inc., Chicago, IL, USA). The Kolmogorov–Smirnov test was performed to assess data distribution, and parametric variables were identified. Baseline data were established using an independent sample *t*-test. Potential statistical differences were evaluated using a repeated-measures two-way ANOVA (time × group). Mean values are reported as standard deviation (SD), and statistical significance was set at $p \leq 0.05$. Cohen's d was calculated by the difference between the means of the groups divided by the combined standard deviation of both groups, considering the sample size. Effect size was considered small for 0.2, moderate for 0.5, and large for 0.8. Finally, an ANCOVA was performed to adjust for the effect of the covariate body mass index, which evaluates the distribution of body mass as a function of body surface area in kg/m$^2$, with the aim of finding significant differences between men and women, adjusting for this covariate. For this purpose, the value of the covariance, regression coefficient, total sum of squares, total sum of squares of the error,

and degrees of freedom were calculated, taking into account the sample size of each of the groups. Finally, the p-statistic and the magnitude of the effect (partial eta squared $\eta^2$) were obtained.

## 3. Results

During SC30, quadriceps activation was higher than that of the hamstrings, except for RMS50 and RMS100, where hamstring activation exceeded that of the quadriceps. The highest H/Q ratio during this directional change was found in the explosive ratio H/Q50 (1.027 ± 2.55). In SC45, all RMS values in the quadriceps were higher than those in the hamstrings. Regarding coactivation ratios, the values were nearly 1 in all cases, although in the traditional ratio H/QMVC, the quadriceps value was slightly higher than that of the hamstrings. Finally, the RMS and ratios were found for SC45cl. In this case, all quadriceps activations were greater than those of the hamstrings. Concerning the ratio, in no case did it exceed the value of 1, indicating higher activation of the anterior musculature compared to the posterior (Table 2).

**Table 2.** Description of EMG in changes of direction.

| | SC30 | | SC45 | | SC45cl | |
|---|---|---|---|---|---|---|
| **Variable** | **Mean** | **SD** | **Mean** | **SD** | **Mean** | **SD** |
| $RMS_Q$_MVC (mV) | 0.6469 | 0.098 | 0.6558 | 0.096 | 0.6682 | 0.108 |
| $RMS_H$_MVC (mV) | 0.6405 | 0.125 | 0.6385 | 0.126 | 0.6471 | 0.121 |
| $RMS_Q$_50 ms (mV) | 0.6448 | 0.098 | 0.6483 | 0.106 | 0.6730 | 0.101 |
| $RMS_H$_50 ms (mV) | 0.6464 | 0.118 | 0.6406 | 0.132 | 0.6506 | 0.124 |
| $RMS_Q$_100 ms (mV) | 0.6466 | 0.099 | 0.6544 | 0.102 | 0.6747 | 0.102 |
| $RMS_H$_100 ms (mV) | 0.6469 | 0.116 | 0.6395 | 0.117 | 0.6484 | 0.123 |
| $RMS_Q$_150 ms (mV) | 0.6428 | 0.099 | 0.6504 | 0.104 | 0.6727 | 0.101 |
| $RMS_H$_150 ms (mV) | 0.6415 | 0.120 | 0.6473 | 0.111 | 0.6478 | 0.126 |
| $RMS_Q$_200 ms (mV) | 0.6402 | 0.099 | 0.6495 | 0.105 | 0.6703 | 0.102 |
| $RMS_H$_200 ms (mV) | 0.6309 | 0.130 | 0.6365 | 0.126 | 0.6506 | 0.130 |
| Ratio_H/$Q_{MVC}$ | 1.011 | 0.252 | 0.9934 | 0.243 | 0.9942 | 0.263 |
| Ratio_H/$Q_{50}$ | 1.027 | 0.255 | 1.0182 | 0.284 | 0.9908 | 0.256 |
| Ratio_H/$Q_{100}$ | 1.025 | 0.256 | 1.0001 | 0.238 | 0.9815 | 0.244 |
| Ratio_H/$Q_{150}$ | 1.025 | 0.285 | 1.0206 | 0.240 | 0.9847 | 0.256 |
| Ratio_H/$Q_{200}$ | 1.009 | 0.273 | 1.0059 | 0.268 | 0.9947 | 0.271 |

Data expressed as mean and SD. RMS (mV) = root mean square; Q = quadriceps; H = hamstrings; MVC = maximum voluntary contraction; 50/100/150/200 (ms) = with respect to the first 50/100/150/200 milliseconds; Ratio_H/Q = ratio of co-activation of hamstrings to quadriceps.

Regarding sex differences (Table 3), although RMSQ showed higher values in men compared to women and RMSH was higher in women than in men, the differences found were not statistically significant in SC30 ($p > 0.05$). However, the Ratio_H/Q50 was significantly higher in women than in men ($p = 0.020$). During SC45, there were significant differences in RMSQ for 50 ms, 100 ms, and 200 ms based on sex, where men showed higher values ($p \leq 0.05$). Although RMSH was not significantly higher in women, the coactivation ratios at 50 ms ($p = 0.044$) and 100 ms ($p = 0.016$) were.

In SC45cl in phase 0, RMSQ was significantly higher in men than in women in all cases ($p \geq 0.05$). In the case of RMSH, we did not find significant differences between sexes. Regarding coactivation, all ratios are significantly higher in women than in men except for the Ratio_H/Q100, which, although higher, was not significant.

Finally, to reflect the possible effect of body mass and height of the subjects, the adjustment of the covariate body mass index was obtained to determine the differences between men and women (Table 4). The data show significant differences in the three changes of direction for the variables analyzed. However, a greater number of significant differences was found in the closed directional changes.

**Table 3.** EMG differences in men and women during directional changes.

| Variables | SC30 | | | | SC45 | | | | SC45cl | | | |
|---|---|---|---|---|---|---|---|---|---|---|---|---|
| | H | M | *p* | d-Cohen | H | M | *p* | d-Cohen | H | M | *p* | d-Cohen |
| $RMS_Q$_MVC (mV) | 0.6534 (0.102) | 0.6254 (0.082) | 0.135 | 0.29 | 0.6641 (0.098) | 0.6285 (0.086) | 0.053 | 0.04 | 0.6807 (0.10) | 0.6276 (0.10) | 0.010 | 0.53 |
| $RMS_H$_MVC (mV) | 0.6353 (0.119) | 0.6574 (0.145) | 0.358 | −0.10 | 0.6351 (0.125) | 0.6498 (0.129) | 0.543 | −0.12 | 0.6426 (0.11) | 0.6618 (0.13) | 0.407 | −0.17 |
| $RMS_Q$_50 ms (mV) | 0.6521 (0.098) | 0.6206 (0.097) | 0.093 | 0.32 | 0.6603 (0.105) | 0.6090 (0.101) | 0.011 | 0.49 | 0.6854 (0.09) | 0.6320 (0.10) | 0.005 | 0.58 |
| $RMS_H$_50 ms (mV) | 0.6378 (0.118) | 0.6744 (0.116) | 0.105 | −0.22 | 0.6373 (0.135) | 0.6513 (0.123) | 0.582 | −0.11 | 0.6479 (0.11) | 0.6597 (0.14) | 0.621 | −0.10 |
| $RMS_Q$_100 ms (mV) | 0.6532 (0.102) | 0.6250 (0.086) | 0.138 | 0.07 | 0.6654 (0.103) | 0.6182 (0.092) | 0.015 | 0.47 | 0.6867 (0.10) | 0.6355 (0.09) | 0.008 | 0.52 |
| $RMS_H$_100 ms (mV) | 0.6400 (0.113) | 0.6694 (0.126) | 0.187 | −0.25 | 0.6345 (0.115) | 0.6559 (0.123) | 0.341 | −0.18 | 0.6477 (0.11) | 0.6509 (0.14) | 0.889 | −0.03 |
| $RMS_Q$_150 ms (mV) | 0.6493 (0.102) | 0.6215 (0.087) | 0.142 | 0.28 | 0.6581 (0.107) | 0.6252 (0.087) | 0.096 | 0.32 | 0.6859 (0.10) | 0.6297 (0.09) | 0.003 | 0.58 |
| $RMS_H$_150 ms (mV) | 0.6364 (0.121) | 0.6583 (0.117) | 0.343 | −0.18 | 0.6435 (0.107) | 0.6597 (0.121) | 0.447 | −0.15 | 0.6428 (0.12) | 0.6640 (0.13) | 0.379 | −0.17 |
| $RMS_Q$_200 ms (mV) | 0.6457 (0.104) | 0.6220 (0.078) | 0.211 | 0.24 | 0.6608 (0.107) | 0.6126 (0.090) | 0.016 | 0.47 | 0.6828 (0.10) | 0.6294 (0.09) | 0.006 | 0.55 |
| $RMS_H$_200 ms (mV) | 0.6236 (0.133) | 0.6550 (0.120) | 0.209 | −0.24 | 0.6304 (0.129) | 0.6565 (0.115) | 0.280 | −0.21 | 0.6442 (0.12) | 0.6715 (0.14) | 0.274 | −0.22 |
| Ratio_H/$Q_{MVC}$ | 0.9942 (0.241) | 1.0687 (0.283) | 0.122 | −0.30 | 0.9758 (0.237) | 1.0517 (0.257) | 0.101 | −0.31 | 0.9658 (0.24) | 1.0858 (0.29) | 0.016 | −0.47 |
| Ratio_H/$Q_{50}$ | 1.0010 (0.243) | 1.1139 (0.274) | 0.020 | −0.45 | 0.9928 (0.274) | 1.1015 (0.302) | 0.044 | −0.39 | 0.9659 (0.23) | 1.0718 (0.29) | 0.030 | −0.43 |
| Ratio_H/$Q_{100}$ | 1.0056 (0.256) | 1.0897 (0.249) | 0.085 | −0.33 | 0.9747 (0.227) | 1.0835 (0.257) | 0.016 | −0.46 | 0.9613 (0.22) | 1.0473 (0.29) | 0.065 | −0.36 |
| Ratio_H/$Q_{150}$ | 1.0088 (0.296) | 1.0796 (0.243) | 0.194 | −0.25 | 1.0036 (0.233) | 1.0763 (0.256) | 0.113 | −0.30 | 0.9557 (0.24) | 1.079 (0.28) | 0.011 | −0.49 |
| Ratio_H/$Q_{200}$ | 0.9921 (0.284) | 1.0669 (0.228) | 0.151 | −0.28 | 0.9792 (0.269) | 1.0935 (0.251) | 0.280 | −0.43 | 0.9647 (0.25) | 1.0923 (0.29) | 0.013 | −0.49 |

Data expressed as mean and SD. $p \leq 0.05$. RMS (mV) = root mean square; Q = quadriceps; H = hamstrings; MVC = maximum voluntary contraction; 50/100/150/200 (ms) = with respect to the first 50/100/150/200 milliseconds; Ratio_H/Q = ratio of co-activation of hamstrings to quadriceps. SC30 = 30° open change of direction. SC45 = 45° open change of direction. SC45cl = 45° closed change of direction towards the same side.

**Table 4.** ANCOVA to determine the effect of the BMI covariate on the differences between men and women.

| Variables | SC30 | | SC45 | | SC45cl | |
|---|---|---|---|---|---|---|
| | *p* | Partial Eta Squared ($\eta^2$) | *p* | Partial Eta Squared ($\eta^2$) | *p* | Partial Eta Squared ($\eta^2$) |
| $RMS_Q$_MVC (mV) | 0.020 | 0.053 | 0.013 | 0.038 | 0.001 | 0.155 |
| $RMS_H$_MVC (mV) | 0.142 | 0.013 | 0.703 | 0.001 | 0.001 | 0.163 |
| $RMS_Q$_50 ms (mV) | 0.039 | 0.026 | 0.467 | 0.003 | 0.001 | 0.275 |

**Table 4.** *Cont.*

| Variables | SC30 | | SC45 | | SC45cl | |
|---|---|---|---|---|---|---|
| | $p$ | Partial Eta Squared ($\eta^2$) | $p$ | Partial Eta Squared ($\eta^2$) | $p$ | Partial Eta Squared ($\eta^2$) |
| RMS$_H$_50 ms (mV) | 0.747 | 0.001 | 0.292 | 0.001 | 0.001 | 0.226 |
| RMS$_Q$_100 ms (mV) | 0.023 | 0.032 | 0.016 | 0.035 | 0.001 | 0.204 |
| RMS$_H$_100 ms (mV) | 0.187 | 0.011 | 0.012 | 0.039 | 0.003 | 0.064 |
| RMS$_Q$_150 ms (mV) | 0.071 | 0.020 | 0.002 | 0.060 | 0.001 | 0.184 |
| RMS$_H$_150 ms (mV) | 0.257 | 0.008 | 0.028 | 0.030 | 0.379 | 0.017 |
| RMS$_Q$_200 ms (mV) | 0.003 | 0.055 | 0.014 | 0.038 | 0.006 | 0.057 |
| RMS$_H$_200 ms (mV) | 0.025 | 0.031 | 0.031 | 0.029 | 0.019 | 0.041 |
| Ratio_H/Q$_{MVC}$ | 0.007 | 0.045 | 0.024 | 0.031 | 0.015 | 0.048 |
| Ratio_H/Q$_{50}$ | 0.005 | 0.048 | 0.005 | 0.048 | 0.031 | 0.038 |
| Ratio_H/Q$_{100}$ | 0.008 | 0.044 | 0.007 | 0.045 | 0.042 | 0.032 |
| Ratio_H/Q$_{150}$ | 0.027 | 0.030 | 0.004 | 0.049 | 0.019 | 0.048 |
| Ratio_H/Q$_{200}$ | 0.025 | 0.031 | 0.031 | 0.028 | 0.026 | 0.036 |

Data expressed as mean and SD. $p \leq 0.05$. $\eta^2$ = effect size. RMS (mV) = root mean square; Q = quadriceps; H = hamstrings; MVC = maximum voluntary contraction; 50/100/150/200 (ms) = with respect to the first 50/100/150/200 milliseconds; Ratio_H/Q = ratio of co-activation of hamstrings to quadriceps. SC30 = 30° open change of direction. SC45 = 45° open change of direction. SC45cl = 45° closed change of direction towards the same side.

## 4. Discussion

The main objective of the current study was to analyze the neuromuscular response through electromyography (EMG) during open and closed directional changes, with a sex-based comparison. Several novel findings emerged from this study. Firstly, we assessed muscle coactivation of the dominant limb using the H/Q ratio, both traditionally (relative to MVC) and in relation to the capacity for rapid or explosive force development, specifically within the first 50, 100, 150, and 200 ms of contraction. This innovative approach has been recently proposed in the scientific literature as a more effective means of determining an increased risk of lower-limb injuries [19,20]. Unlike the traditional MVC protocol, which shows peak force values around 5 s, analyzing the H/Q ratio with respect to the first 200 ms aligns with the time period when the risk of injury is at its peak [21]. In this regard, the study provides a novel perspective on the neuromuscular response using EMG with traditional H/QMVC and explosive H/Q ratios.

To analyze participants' EMG response, we assessed the root mean square (RMS) of the rectus femoris as a representative of the quadriceps and the midline of the semimembranosus biceps femoris as a representative of the hamstrings, following the protocol of Lim et al. [22]. The coactivation ratio was obtained by dividing the H/Q activation [23]. As indicated by Dallinga et al. [24], the H/Q ratio stands out as one of the best indicators of potential knee injuries. Typically, an imbalance in the activation of the posterior musculature relative to the anterior is a decisive factor in injury onset [25]. Detailed analysis of anterior and posterior musculature activation revealed that RMSQ was generally higher than RMSH in all directional changes, except for RMS50 and RMS100, where posterior musculature activation was slightly higher than anterior during SC30. Increased activation of the anterior musculature may signify greater issues with the anterior cruciate ligament (ACL), especially at knee flexion angles ranging from total extension to 30–45° [26]. Our kinematic results indicated knee flexion ranging from 0° (total extension) to 45° in all directional changes, suggesting that this anterior musculature activation, accompanied by these knee flexion angles, could lead to a notable increase in the risk of knee joint injuries. However, superior activation of the anterior musculature over the posterior seems insufficient [27]. Undoubtedly, the phenomenon best describing the risk of injury is the timing of muscle

activation, as it corresponds to how activations of the posterior musculature relate to the anterior [28,29].

Our results demonstrate a coactivation ratio very close to 1 in all directional changes, a key aspect indicating a protective effect in terms of the muscle control participants exhibited during directional changes. However, 40% of subjects showed coactivation ratios of 0.8 or lower during these movements, describing worse coactivation of the posterior musculature relative to the anterior and thus increasing the risk of injury (Cameron et al., 2003).

Furthermore, our findings indicate that there were no appreciable variations in the activation of the anterior and posterior thigh muscles during the SC30 between men and women. This aligns with the findings of Cowan and Crossley [30], who demonstrated no significant sex differences in the onset of EMG activity in the major thigh muscles. Nevertheless, when adjusted for the covariate BMI, it was observed that BMI did influence the sex differences, especially in the activation of the anterior thigh muscles. However, when examining explosive ratios in detail, it is interesting to note that in the H/Q50 ratio, women significantly surpassed men, surpassing the H/Q ratio of 1.1. A possible explanation could be the overactivation of the posterior thigh musculature due to uncontrolled motor unit recruitment, a characteristic often observed in women [26]. This may signify a negative factor contributing to an increased risk of injuries since, as suggested by Ghering et al. [31], simultaneous activation of both anterior and posterior thigh musculature entails better knee joint control. During the SC45, not only did women exhibit a significantly higher H/Q50 ratio than men, but this phenomenon also occurred in the H/Q100 ratio.

**5. Conclusions**

This study, aimed at providing a comprehensive analysis of the neuromuscular response using EMG during directional changes, highlighted the significance of an innovative approach to assessing the coactivation ratio (explosive ratio) in addition to the traditional ratio. In its association with an increased risk of injury, the early moments of this coactivation appear to be linked to the likelihood of injury. Simultaneously, activation of the anterior thigh musculature was higher than the posterior in all directional changes, except in some open changes of direction (concretely, in SC30). However, concerning the EMG response based on the ratio, women exhibited poorer coactivation compared to men, potentially explaining an increased risk of knee injury during both open and closed directional changes.

**Author Contributions:** Research concept and study design (A.F.-M. and B.S.), literature review (A.F.-M. and N.B.), data collection (A.F.-M.), data analysis and interpretation (A.F.-M., B.S. and J.A.P.), statistical analyses (A.F.-M.), writing of the manuscript (A.F.-M. and B.S.), reviewing/editing a draft of the manuscript (J.A.P. and N.B.). All authors have read and agreed to the published version of the manuscript.

**Funding:** This research received no external funding.

**Institutional Review Board Statement:** The study was conducted in accordance with the Declaration of Helsinki and approved by the Ethics Committee of University of Seville (protocol code: 29062012; date of approval: 29 June 2012).

**Informed Consent Statement:** Informed consent was obtained from all subjects involved in the study.

**Data Availability Statement:** The original contributions presented in the study are included in the article, further inquiries can be directed to the corresponding author.

**Acknowledgments:** The authors would like to thank all those who made this study possible.

**Conflicts of Interest:** The authors declare no conflicts of interest.

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
