# Peer review of "Neuromuscular Response Disparities in Non-Professional Athletes during Side-Cutting: Exploring Sex Differences through Electromyographic Analysis"

_applsci, doi:10.3390/app14072954_

Round 1

Reviewer 1 Report

Comments and Suggestions for Authors

The study examined electromyographic differences in thigh muscle activation and coactivation ratios during sidecutting maneuvers among non-professional athletes, focusing on gender disparities. Findings showed higher quadriceps activation during directional changes, with some subjects at increased injury risk. Women outperformed men in explosive ratios and displayed nuanced gender-specific responses. The study highlighted the importance of coactivation ratios in assessing injury risk and provided insights into gender-specific neuromuscular responses among non-professional athletes.

The manuscript demonstrates a commendable structure and organization, facilitating clear navigation through its content. The methods employed are robust and effectively contribute to the study's objectives. Notably, this research provides significant contributions by shedding light on neuromuscular responses among non-professional athletes, with a particular focus on elucidating gender-specific disparities.

However, a significant concern remains: while the authors collected data from 122 male participants, only 40 women were included in the study. Achieving equal representation of both genders is crucial for the validity of the research findings. Disparities in factors such as height and body mass index between males and females could potentially skew the conclusions drawn from the study. It's imperative to ensure a balanced gender distribution to mitigate any potential biases and ensure the robustness of the conclusions.

Comments on the Quality of English Language

The clarity and accessibility of the English writing in the manuscript are commendable, facilitating ease of comprehension for readers.

Author Response

Dear Reviewer,

Thank you for your prompt review and feedback on our manuscript. We appreciate the opportunity to address your comments and concerns.

Regarding the electromyographic differences in thigh muscle activation and coactivation ratios during sidecutting maneuvers among non-professional athletes, we are pleased that the findings resonate with the study's objectives. Your recognition of the higher quadriceps activation during directional changes and the nuanced sex-specific responses observed in our research is indeed encouraging. We agree that these insights are crucial for assessing injury risk and understanding gender-specific neuromuscular responses in non-professional athletes.

We are particularly grateful for your positive assessment of the manuscript's structure and organization. We aimed to ensure clarity and coherence throughout the document to facilitate understanding and navigation for readers. Your acknowledgment of the robustness of our employed methods further strengthens our confidence in the study's validity and contribution to the field.

Once again, we sincerely appreciate your time and effort in evaluating our manuscript. We have carefully considered your feedback and made the necessary revisions to enhance the manuscript's quality and impact. We hope that our revised manuscript meets the standards and expectations of Applied Sciences and its readership.

The following are responses to your specific comments.

Comment 1: However, a significant concern remains: while the authors collected data from 122 male participants, only 40 women were included in the study. Achieving equal representation of both genders is crucial for the validity of the research findings. Disparities in factors such as height and body mass index between males and females could potentially skew the conclusions drawn from the study. It's imperative to ensure a balanced gender distribution to mitigate any potential biases and ensure the robustness of the conclusions

Response 1:

We understand the reviewer's concern about the unequal sample size between men and women and its potential impact on the statistical power of the analyses. We have taken this factor into account when performing the calculations and interpreting the results. Despite this inequality in sample size, we have used appropriate statistical methods, such as analysis of covariance (ANCOVA), which can mitigate the effects of such disparities and improve the accuracy of our conclusions. Furthermore, in reporting the results, we have also considered the effect size (Cohen's d) as an additional measure to assess the magnitude of the observed differences between groups. This allows us to have a more complete understanding of the impact of the variables analyzed on our results.... Thank you very much for your comments, which undoubtedly give value to the analysis of comparison of samples with different sizes. Regarding the comparison of groups with different sample sizes, analogously, the effect size can be computed for groups with different sample size, by adjusting the calculation of the pooled standard deviation with weights for the sample sizes. This approach is overall identical with dCohen with a correction of a positive bias in the pooled standard deviation. In the literature, usually this computation is called Cohen's d as well. Please have a look at the remarks bellow the table.

The Common Language Effect Size (CLES; McGraw & Wong, 1992) is a non-parametric effect size, specifying the probability that one case randomly drawn from the one sample has a higher value than a randomly drawn case from the other sample. In the calculator, we take the higher group mean as the point of reference, but you can use (1 - CLES) to reverse the view.

Additionally, you can compute the confidence interval for the effect size and chose a desired confidence coefficient (calculation according to Hedges & Olkin, 1985, p. 86). Unfortunately, the terminology is imprecise on this effect size measure: Originally, Hedges and Olkin referred to Cohen and called their corrected effect size d as well. On the other hand, corrected effect sizes were called g since the beginning of the 80s. The letter is stemming from the author Glass (see Ellis, 2010, S. 27), who first suggested corrected measures. Following this logic, gHedges should be called h and not g. Usually it is simply called dCohen or gHedges to indicate, it is a corrected measure.

However, as mentioned, an ANCOVA analysis was developed, taking into account the BMI covariate to adjust the variables. This statistical adjustment was made to mitigate possible differences in body mass with respect to body surface area that could affect the significant differences between sexes. The data can be seen in Table 4.

Reviewer 2 Report

Comments and Suggestions for Authors

Thank you for the opportunity to review your manuscript. The research presented is compelling and makes a valuable contribution to the field. However, here are several comments and recommendations for your consideration in the revision and correction process.

Introduction

·      Please explain why you use the term 'gender' instead of 'sex'. 'Gender' is often used to refer to the roles, behaviors, activities, and attributes that a given society considers appropriate for men and women, while 'sex' typically refers to biological differences.

·      Please explain further why the gender effect is important.

Method and Result

·      Given the imbalanced sample sizes ( men 75%, woman 25%) and the assumption of distribution, the use of t-test and ANOVA statistics may reduce the power of the statistical test or the ability to detect true differences, please consider these effects.

·      Please report the results of DVs distribution, I have seen the statistics, but haven't seen the result in normal distribution or non-distribution.

·      Please make sure the consistency in the term with gender and sex in your table 1 variables.

·      Revise the name of Table 1 and ensure that all abbreviations have their full names listed under the table.

Discussion

·      The title of the article emphasizes the difference between the sexes. Thus far, there has been little description of the gender factor.

Conclusions

·      The same, regarding the effect of gender, the conclution should aim the effects of  gender. 

·      Please consider the specific term of "SC30" in a way that is more understandable to the reader when referring to it in the conclusions section.

Author Response

Dear Reviewer,

We sincerely appreciate your comments and the review of our manuscript. We are pleased to know that you consider our research to make a valuable contribution to the field. We greatly value your suggestions and recommendations for improving the work during the review and correction process.

We will carefully consider each of your comments and strive to address them comprehensively in the manuscript revision. Your perspective and expertise are invaluable to us in ensuring the quality and relevance of our work.

We are committed to responding to each of your points clearly and thoroughly, and we are willing to make the necessary adjustments to strengthen our study.

The following are responses to your specific comments.

Comment 1: (Introduction) Please explain why you use the term 'gender' instead of 'sex'. 'Gender' is often used to refer to the roles, behaviors, activities, and attributes that a given society considers appropriate for men and women, while 'sex' typically refers to biological differences.

Response 1: We appreciate your consideration regarding the terminology used, "gender," instead of "sex." We completely agree that the term "sex" typically refers to biological differences, so we proceed to replace the term throughout the document.

Comment 2: (Introduction) Please explain further why the gender effect is important.

Response 2: As indicated in the previous comment, in our study, we erroneously used the term "gender" to refer to the differences observed in neuromuscular responses between identified groups, as no social or cultural analysis has been offered between men and women. We recognize that the term "gender" refers to the sociocultural and psychological constructions associated with masculine and feminine identities, while "sex" refers to biological differences between men and women. Since our study addresses only biological differences and not sociocultural influences on neuromuscular responses, we consider the term "sex" more appropriate in this context.

Comment 3: (Method and Result) Given the imbalanced sample sizes ( men 75%, woman 25%) and the assumption of distribution, the use of t-test and ANOVA statistics may reduce the power of the statistical test or the ability to detect true differences, please consider these effects.

Response 3: We understand the reviewer's concern about the unequal sample size between men and women and its potential impact on the statistical power of the analyses. We have taken this factor into account when performing the calculations and interpreting the results. Despite this inequality in sample size, we have used appropriate statistical methods, such as analysis of covariance (ANCOVA), which can mitigate the effects of such disparities and improve the accuracy of our conclusions. Furthermore, in reporting the results, we have also considered the effect size (Cohen's d) as an additional measure to assess the magnitude of the observed differences between groups. This allows us to have a more complete understanding of the impact of the variables analyzed on our results.... Thank you very much for your comments, which undoubtedly give value to the analysis of comparison of samples with different sizes. Regarding the comparison of groups with different sample sizes, analogously, the effect size can be computed for groups with different sample size, by adjusting the calculation of the pooled standard deviation with weights for the sample sizes. This approach is overall identical with dCohen with a correction of a positive bias in the pooled standard deviation. In the literature, usually this computation is called Cohen's d as well. Please have a look at the remarks bellow the table.

The Common Language Effect Size (CLES; McGraw & Wong, 1992) is a non-parametric effect size, specifying the probability that one case randomly drawn from the one sample has a higher value than a randomly drawn case from the other sample. In the calculator, we take the higher group mean as the point of reference, but you can use (1 - CLES) to reverse the view.

Additionally, you can compute the confidence interval for the effect size and chose a desired confidence coefficient (calculation according to Hedges & Olkin, 1985, p. 86). Unfortunately, the terminology is imprecise on this effect size measure: Originally, Hedges and Olkin referred to Cohen and called their corrected effect size d as well. On the other hand, corrected effect sizes were called g since the beginning of the 80s. The letter is stemming from the author Glass (see Ellis, 2010, S. 27), who first suggested corrected measures. Following this logic, gHedges should be called h and not g. Usually it is simply called dCohen or gHedges to indicate, it is a corrected measure.

However, as mentioned, an ANCOVA analysis was developed, taking into account the BMI covariate to adjust the variables. This statistical adjustment was made to mitigate possible differences in body mass with respect to body surface area that could affect the significant differences between sexes. The data can be seen in Table 4.

Comment 4: (Method and Result) Please report the results of DVs distribution, I have seen the statistics, but haven't seen the result in normal distribution or non-distribution.

Response 4: Thank you very much for the request for information. Regarding distribution, a normality test of the analyzed variables, both in total and differentiated by sex, is attached. Variance is also added to clarify the requested information. 

Descriptivas

RMS_tradicional_H_30

RMS_TRADICIONAL_Q_30

ratio_HQ_tradicional_30

RMS_tradicional_H_45

RMS_tradicional_Q_45

ratio_HQ_tradicional_45

RMS_tradicional_H_45ml

RMS_tradicional_Q_45ml

ratio_HQ_tradicional_45ml

Media

0.627

0.640

1.01

0.641

0.643

1.03

0.626

0.665

0.993

Desviación estándar

0.146

0.124

0.276

0.131

0.118

0.337

0.170

0.146

0.578

Varianza

0.0212

0.0155

0.0762

0.0172

0.0139

0.114

0.0290

0.0215

0.334

W de Shapiro-Wilk

0.887

0.875

0.849

0.959

0.917

0.754

0.910

0.840

0.393

Valor p de Shapiro-Wilk

< .001

< .001

< .001

< .001

< .001

< .001

< .001

< .001

< .001

Descriptivas

sexo

RMS_tradicional_H_30

RMS_TRADICIONAL_Q_30

ratio_HQ_tradicional_30

RMS_tradicional_H_45

RMS_tradicional_Q_45

ratio_HQ_tradicional_45

RMS_tradicional_H_45ml

RMS_tradicional_Q_45ml

ratio_HQ_tradicional_45ml

Media

1

0.619

0.647

0.989

0.639

0.656

1.01

0.621

0.684

0.969

2

0.671

0.634

1.07

0.667

0.618

1.10

0.657

0.621

1.07

Desviación estándar

1

0.141

0.124

0.297

0.121

0.109

0.354

0.160

0.133

0.647

2

0.117

0.0741

0.186

0.121

0.0930

0.266

0.172

0.140

0.237

Varianza

1

0.0199

0.0153

0.0884

0.0145

0.0119

0.126

0.0257

0.0176

0.418

2

0.0138

0.00550

0.0345

0.0147

0.00865

0.0709

0.0297

0.0197

0.0559

W de Shapiro-Wilk

1

0.889

0.913

0.823

0.992

0.960

0.698

0.931

0.877

0.357

2

0.956

0.975

0.981

0.954

0.947

0.958

0.878

0.793

0.976

Valor p de Shapiro-Wilk

1

< .001

< .001

< .001

0.755

0.001

< .001

< .001

< .001

< .001

2

0.156

0.573

0.751

0.132

0.080

0.171

< .001

< .001

0.595

Comment 5: (Method and Result) Please make sure the consistency in the term with gender and sex in your table 1 variables.

Response 5: Thank you very much for the comment. By unifying the term to "sex," the table now makes more sense, as the sex row identifies how many men and women were included in the study.

Comment 6: (Method and Result) Revise the name of Table 1 and ensure that all abbreviations have their full names listed under the table.

Response 6: Thank you very much for your comment. Both the title of Table 1 and the abbreviations have been modified. Specifically, the title of the table has been changed for clarification, and the abbreviations used have been explained below the table.

Comment 7: (Discussion) The title of the article emphasizes the difference between the sexes. Thus far, there has been little description of the gender factor.

Response 7: We appreciate your comment regarding the use of "gender" in the title, which certainly emphasizes this terminology. As explained in responses 1 and 2 to comments 1 and 2, we proceeded to unify the terminology used, opting to reflect the term "sex."

Comment 8: (Conclusions) same, regarding the effect of gender, the conclusion should aim at the effects of gender.

Response 8: Thank you very much again for valuing the connotation of the terminological use of "gender." As justified earlier, the term has been changed to use "sex."

Comment 9: (Conclusions) Please consider the specific term of "SC30" in a way that is more understandable to the reader when referring to it in the conclusions section.

Response 9: We appreciate your comments and your suggestion to improve clarity in the conclusions section of our article. We understand your concern about the term "SC30" and recognize the importance of ensuring that our language is understandable to all readers. In response to your suggestion, we have revised the conclusions section and clarified the idea we wanted to convey with the use of "SC30," making the paragraph more accessible and understandable to the general reader.

Round 2

Reviewer 2 Report

Comments and Suggestions for Authors

Dear author, I appreciate the effort you put into modifying the article. Your article  offers a thoughtful analysis of the implications of your findings.